# Does Inspiratory Muscle Training Affect Static Balance in Soccer Players? A Pilot Randomized Controlled Clinical Trial

**DOI:** 10.3390/healthcare11020262

**Published:** 2023-01-14

**Authors:** Silvana Loana de Oliveira-Sousa, Martha Cecilia León-Garzón, Mariano Gacto-Sánchez, Alfonso Javier Ibáñez-Vera, Luis Espejo-Antúnez, Felipe León-Morillas

**Affiliations:** 1Department of Physiotherapy, University of Murcia, 30003 Murcia, Spain; 2Department of Physiotherapy, Jerónimos Campus, 135. Catholic University of Murcia UCAM, Guadalupe, 30107 Murcia, Spain; 3Department of Health Sciences, Campus de las Lagunillas, University of Jaén, 23071 Jaén, Spain; 4Department of Medical-Surgical Therapeutics, Faculty of Medicine and Health Sciences, University of Extremadura, 06006 Badajoz, Spain

**Keywords:** inspiratory muscle training, static balance, center of pressure, soccer players

## Abstract

Inspiratory muscle training (IMT) is effective in improving postural stability and balance in different clinical populations. However, there is no evidence of these effects in soccer players. A single-blind, two-arm (1:1), randomized, placebo-controlled pilot study on 14 soccer players was performed with the main aim of assessing the effect of IMT on static balance, and secondarily, of examining changes in the respiratory muscle function. The experimental group (EG) received an IMT program with progressive intensity, from 20% to 80%, of the maximal inspiratory pressure (MIP). The sham group (SG) performed the same program with a fixed load of 20% of the MIP. Static balance and respiratory muscle function variables were assessed. A two-factor analysis of variance for repeated measures was used to assess differences after training. Statistical significance was set at *p* < 0.05. Significant increases were observed in the EG on length of sway under eyes open (from 2904.8 ± 640.0 to 3522.4 ± 509.0 mm, *p* = 0.012) and eyes closed (from 3166.2 ± 641.3 to 4173.3 ± 390.8 mm, *p* = 0.004). A significant increase in the maximal voluntary ventilation was observed for both groups (EG *p* = 0.005; SG *p* = 0.000). No significant differences existed between the groups. IMT did not improve the static balance in a sample of soccer players. Conducting a high-scale study is feasible and could refine the results and conclusions stemming from the current pilot study.

## 1. Introduction

Postural stability is an essential motor skill in soccer, and postural talent might be considered an indicator of performance [1]. Soccer is a sport with high physical demands. Some of them involve the respiratory muscles, which can significantly contribute to the limitation of exercise, due to muscle fatigue and the effects on blood flow in the limbs that perform physical work. In addition, postural stability can be altered by respiratory movements [2,3,4] since these interfere with the biomechanics of the torso and postural sway, reducing these aspects under apnea conditions [5] and increasing them with hyperventilation [6]. During a soccer match, players must use their motor skills and control their posture, while also gathering visual information about other team members as well as opponents. Due to the conditions of the game itself, it is of paramount importance for players to have a good balance to control, pass or shoot the ball [7,8].

Inspiratory muscle training (IMT) has been shown to be effective in improving variables related to postural stability in a wide variety of clinical and healthy populations [9,10,11,12,13,14]. These effects are supported by empirical data on the assistance of the respiratory muscles in trunk stabilization: the diaphragm coordinates with the abdominal muscles, therefore generating a hydraulic effect in the abdominal cavity which, in turn, assists the spinal stabilization by stiffening the lumbar spine through increased intra-abdominal pressure [15,16,17,18]. This action assists in maintaining postural stability in situations where external forces (i.e., rapid movement of the upper limb) destabilize the spine, and during reactive and dynamic tasks [16]. Inspiratory muscle training increases the efficiency of diaphragmatic phasic contractions, and the ability to increase intraabdominal pressure, improving balance abilities [14].

Under normal conditions, the respiratory and postural functions of the diaphragm can be coordinated when the stability of the trunk is challenged by repeated rapid movements of the limbs [16]. However, this coordination can be affected by several factors, such as increased respiratory demand due to exercise or weakness of the respiratory muscles [19]. During a soccer game, the respiratory demand is increased, due to high-intensity intermittent activity that requires performers to undertake regular repeated sprints across a 90-min game, and where the sustained level of effort approaches the anaerobic threshold (75% of maximal effort).

In a recent case–control study, we found that greater values of inspiratory muscle strength are associated with shorter path length and less lateral displacement in the closed eyes condition in a sample of soccer players [20]. This association may lead to the hypothesis that a specific IMT program in soccer players could improve their postural stability. Data supporting or rejecting this hypothesis are, however, scarce in these populations. In previous literature reviews encompassing different types of players and specifically involving soccer players, none of the included randomized controlled trials (RCTs) examined variables related to balance [21,22]. Thus, the primary objective of this study was to assess the effect of IMT on static balance, and secondarily, to examine changes in the respiratory muscle function.

## 2. Materials and Methods

### 2.1. Study Design

A single-blind (evaluators) pilot randomized controlled clinical trial was carried out in a sample of soccer players from the U23 soccer team of the Catholic University of Murcia (UCAM). The study was developed from February to May 2018. This study has followed the ethical principles of the Declaration of Helsinki, was approved by the corresponding Ethics Committee and has been registered in ClinicalTrials.gov under the code NCT03383900. The CONSORT statements for conducting and reporting this randomized controlled trial were also followed. All subjects agreed to take part in the study and signed the corresponding informed consent.

### 2.2. Participants

Soccer players from the UCAM U23 men’s soccer team during the season 2017–2018 were included in this study. Inclusion criteria included attending training sessions regularly, alongside participating in over 80% of the games from the start of the season until the first measurement. Exclusion criteria included severe musculoskeletal injuries and lung-based pathologies. Informed consent forms were signed by every player included in the study.

All the athletes were requested to continue their current training regime, with no changes in volume or training intensity.

### 2.3. Randomization and Masking

After signing an informed consent form, participants were randomly assigned (1:1) to an experimental group (EG) or sham group (SG), using a computer-generated random number table. The randomization code was performed by a physiotherapist who did not participate in the measurements nor in the intervention. Until the end of training, these codes were known only by the physiotherapist responsible for randomization. Assessors were also blinded to the participants’ intervention assignment.

### 2.4. Interventions

#### 2.4.1. Experimental Group (EG)

The EG players performed IMT with a Power Breathe device model Heavy Resistance—HR plus (Power Breathe International Ltd., Southam, Warwickshire; England, UK). The IMT consisted of daily sessions (3 sets of 15 repetitions—inspirations), 6 days a week, for a period of 8 weeks. The IMT was performed prior to routine training in the changing rooms of the soccer field and was supervised by a physiotherapist. Whenever players did not train on the field and/or when a player did not attend the training session, they had to send a video performing the IMT at home to a WhatsApp group in which the physiotherapist supervised the activity. The EG players started by breathing at a resistance equal to 20% of their maximum inspiratory pressure (MIP) for one week. The load was then increased incrementally, 5–10% each session, to reach 80% of their MIP at the end of the first month. After week 4, the MIP value was reevaluated and a resistance corresponding to 80% of this new MIP value was used throughout the second month, therefore achieving a total of 8 weeks of training [23,24,25,26].

#### 2.4.2. Sham Group (SG)

Players from the SG performed IMT with Power Breathe. The training program was the same as that of the EG in terms of frequency and duration, whereas the intensity was 20% of the MIP throughout the training period.

### 2.5. Measurements

Data were collected by two physiotherapists (FLM, SLOS) experienced in the field of spirometry. Data collection took place at the same time each day, in the morning timeframe, in the same facility (Physiotherapy practice room from the UCAM), and under similar temperature conditions (24–25 °C).

#### 2.5.1. Static Balance

Static balance was measured through the analysis of stabilometric variables on a balance platform (Free Step platform, Rome, Italy), with an active surface of 400 × 400 mm and 8 mm thickness, and Free-Step v.1.0.3 software (Rome, Italy). Four stabilometric variables were examined in the bipodal eyes open (BOA) and bipodal eyes closed (BOC) conditions: length of sway (LS), surface of the ellipse (SE), lateral axis (DX) and antero-posterior axis (DY). The measurement procedure was performed according to a previously published protocol [20,27]. The players had to stand on the platform with bipodal support (45°) for 90 s with eyes open and 90 s with eyes closed. During the measurement under the open-eyes condition, the players had to keep their gaze on a fixed point that coincided with the height of their eyes and was about 2.5 m away from the platform.

#### 2.5.2. Respiratory Muscle Function

To assess respiratory muscle function, both the inspiratory muscle strength and respiratory endurance were examined. Inspiratory muscle strength was measured indirectly, through the maximal inspiratory pressure (MIP), obtaining values of both the absolute value of MIP and the predictive value in % of MIP (values of normality in Caucasian population) [28]. A maximum inspiratory mouth pressure monitor Datospir Touch (Sibelmed, Barcelona, Spain) was used. With the individuals in a sitting position and wearing a nose clip, they were asked to perform a maximal expiration (close to the residual volume), followed by a maximal inspiration (close to the total lung capacity). The measurement was carried out by a previously trained physiotherapist, performing a maximum of eight measurements, until reaching three measurements that obtained a difference of ≤10%, thus selecting the best result according to the protocol established by the Spanish Society of Pneumology and Thoracic Surgery (SEPAR) [29].

Respiratory muscular endurance was obtained by measuring the maximal voluntary ventilation (MVV): it consists in mobilizing the maximum amount of air in a short period of time (15 s) with a respiratory rate beyond 80 breaths per minute, according to the criteria established by the SEPAR [29]. Subjects in a sitting position and wearing a nose clip were asked to breathe as fast and deep as possible for 15s. Three maneuvers were carried out with 1-min rest between maneuvers, and the best value among the three maneuvers was chosen. Data on both MVV in liters (MVV l, MVV% predict) and number of breaths per minute (MVV bpm) were recorded. These variables were measured with a digital spirometer (Datospir Touch, Sibelmed, Barcelona, Spain). In addition, data on age, weight and height were collected.

### 2.6. Data Analysis

Considering the required sample size, a calculation through G* Power software (ver. 3.1.9.7; Heinrich-Heine-Universität Düsseldorf, Düsseldorf, Germany; http://www.gpower.hhu.de/ -accessed on 26 November 2022-) was performed using the following parameters: one tail, large effect size d = 0.8, alpha = 0.05, statistical power = 0.80 and a 1:1 allocation ratio. The total sample size would consist of 42 subjects (i.e., 21 individuals per group); therefore, our study corresponds to a pilot study, since a total of 18 subjects were initially recruited and 14 completed the study (EG = 7; SG = 7). The perception that pilot trials are simply a casual prelude to a larger trial may somehow threaten the rigor with which they are implemented, and such an approach to pilot trial design and implementation runs the risk of providing misleading results, as stated by Arnold et al. [30]. For this reason, the current study followed the recommendations developed by Thabane et al. [31] for reporting the results of pilot studies.

All results are presented as mean (SD) when applicable. Data were tested for normality using the Shapiro–Wilk test (considering the final size of 14 subjects). A comparison of means for independent samples was carried out, using Student’s *t*-test to determine possible differences between the experimental and sham groups at baseline. Additionally, for inspiratory muscle strength at baseline, a descriptive analysis was performed, based on the predictive values of normality. A two-factor analysis of variance for repeated measures was used to observe differences after training, both within the same group at different times (pre-intervention and post-intervention), and between groups (experimental versus sham) for each one of the variables of interest. Mean differences between groups were also calculated for the variables approached. Statistical significance was set at *p* < 0.05. The effect size was calculated using partial eta-squared and interpreted as small (η^2^ > 0.01), medium (η^2^ > 0.06), or large (η^2^ > 0.14) [32]. All analyses were performed using SPSS software version 22.0 (IBM, Armonk, NY, USA).

## 3. Results

### 3.1. Selection Process

The UCAM U23 soccer team was composed of a total of twenty-two players. The initial sample consisted of eighteen players, since four of them did not meet the inclusion criteria. The random assignment performed led to a distribution of nine players for the EG and nine for the SG. During the study, there were sample losses. Finally, fourteen players completed the study (EG = 7; SG = 7) (Figure 1).

### 3.2. Characteristics of the Participants

Table 1 shows the sociodemographic and anthropometric characteristics alongside the values corresponding to the respiratory muscle function and the static balance in both groups. Groups were homogeneous at baseline, with no statistically significant differences across variables.

### 3.3. Static Balance

Table 2 shows the results of the analysis of stabilometric variables under the condition of eyes open and eyes closed with bipodal support. Significant changes were found in the EG on the variable length of sway under the condition of eyes open (BOA LS) with an increase from an initial value of 2904.8 ± 640.0 to 3522.4 ± 509.0 mm (*p* = 0.012). No significant changes were found in the other stabilometric variables under the open-eyes condition analyzed. Regarding the closed-eyes condition, the length of sway (BOC LS) also showed significant changes in the EG with a baseline value of 3166.2 ± 641.3 mm and a post-intervention value of 4173.3 ± 390.8 mm (*p* = 0.004). For the rest of the variables under the aforementioned condition, there were no significant changes in any of the groups.

In relation to the differences between the groups, no significant changes were found.

### 3.4. Respiratory Muscle Function

Regarding the inspiratory muscle strength (MIP% predictive) at baseline, the mean predictive percentage of the fourteen players (total sample) was 108.1% (96.30–120.0; 95% CI). A total of 25% of the players presented average values below their normality, that is, below 100% of the predictive one. On the other hand, 50% showed predictive values on the borderline value of normality, with an average of 102.4% (Figure 2).

After the training period (8 weeks), the EG showed an increase from 161.5 ± 31.1 to 184.4 ± 21.5 cm H_2_O with a significance level of *p* = 0.076. Regarding the SG, it ranged from 175.4 ± 30.2 to 176.0 ± 16.4 cm H_2_O (*p* = 0.951) (Table 2). In terms of predictive values, the EG improved from 104.8 ± 21.4 to 119.5 ± 14.5 cmH2O, while the change in the SG was negligible, as well as the effect size (ŋ^2^), which was very large in the EG compared to that of the SG (0.432 vs. 0.000, respectively).

Regarding MVV (Table 3), increases in the number of breaths per minute with statistically significant changes were observed in both groups. The EG obtained an initial value of 89.8 ± 15.3 bpm and post-intervention of 143.2 ± 21.9 bpm with *p* = 0.005, while the SG obtained a pre-intervention result of 86.0 ± 21.5 bpm and post-intervention of 147.8 ± 33.4 bpm with *p* = 0.000. Differences between groups were non-significant. Regarding MVV in liters and in % of the predictive value, none of the groups presented significant increases and no significant differences between groups were found either.

## 4. Discussion

The main objective of this study was to determine whether an 8-week inspiratory muscle-specific training program can improve static balance in soccer players. The results revealed no improvement in any of the static balance variables studied. Despite the fact that significant changes in the variable length of sway were stated concerning the experimental group, these changes corresponded to an increase after the training period, unexpectedly and counterintuitively. Secondarily, we observed significant increases in the maximum voluntary ventilation (breaths per minute) in both groups and an important increase, bordering statistical significance, in the inspiratory muscle strength in the IMT group. However, no significant differences on any of the variables examined were found between the groups.

Across all the stabilometric variables examined, significant changes were found only on the length of sway for the IMT group. Surprisingly, this group showed an increase in the length of the postural oscillations, both in the eyes open and eyes closed conditions (although not reaching statistically significant differences when compared to the placebo group). These results are contradictory to previous studies conducted in other healthy populations [13,14]. Rodrigues et al. [13] investigated the effect of a 4-week IMT program on postural balance responses during orthostatic stress in healthy elderly women and observed a reduction in the CoP distance and velocity in the IMT group compared to the sham group.

Unfortunately, the scarcity of data on soccer players hinders a possible comparison with other RCTs carried out on soccer players. In our recent case–control study, we observed that the inspiratory muscle strength was negatively correlated with sway oscillations [20]. This fact led us to the hypothesis that an IMT program could reduce the oscillations of the center of pressure in those players. Two possible reasons could explain the contradictory response observed among our sample of players: On the one hand, the postural strategies adopted by athletes may differ considerably from those of patients; therefore, the response to an IMT may be subsequently different. On the other hand, the increase in the length of sway can perhaps be understood from an ecological-approach perspective, since some authors affirm that the postural oscillation based on the support of asymptomatic young subjects, generated by postural fluctuations, can provide sensory exploratory information about how the body itself interacts with the environment. These fluctuations generate shifts in the center of pressures and depend on the postural patterns of each individual. Therefore, from this perspective, the observed results could be interpreted as an increase in the oscillation to coordinate the postural fluctuations and to have a greater adaptability to the environment [27,33,34].

In spite of the growing interest concerning the role played by IMT in postural control and balance, most of the underlying mechanisms that may explain this relationship remain uncertain. The improvement in the inspiratory muscle strength (MIP) does not seem to entail a better postural control per se. The enhancement of MIP can have an impact on other variables which, in turn, can alter postural control. For example, improvement in the diaphragmatic strength can affect the functionality of the core as a whole, which, in turn, improves trunk stability and balance.

Regarding the changes on variables concerning the respiratory muscle function, the players in the IMT group showed an important increase (14% vs. 0.1% in SG) in the inspiratory muscle strength. However, this increase fell short of being statistically significant. Furthermore, both groups increased their number of breaths per minute in the maximal voluntary ventilation test, but there was no improvement in the number of liters of air mobilized per minute. The characteristics of the training protocol and the device used may have affected the results. In our study, we performed a semi-supervised intervention protocol, of one daily session, while most authors used two-session-per-day protocols to train athletes [23,24,25,26,34,35]. The frequency of one session per day could, therefore, be insufficient to achieve important changes across healthy populations. On another note, an inspiratory threshold load device was used, which may have little effect on the amount of air mobilized and, therefore, on the maximum minute ventilation when compared to devices for voluntary isocapnic hyperpnea [36]. Finally, the results of the evaluation of the respiratory muscle strength at baseline confirm that these players present a respiratory condition not very different from other non-athletic subjects [37]. Due to the high demand that these players present and the subsequent probability of fatigue of untrained respiratory muscles, we believe that monitoring and training these muscles is of paramount importance.

From a methodological perspective, the aim of the current pilot study was to analyze the feasibility of a larger-scale clinical trial through the analysis of its validity and feasibility. Thus, the randomization and blinding, the acceptability of the intervention and the selection of the primary outcome revealed a high level of feasibility of the study protocol [38]. Only the recruitment and consent aspect’s results were somehow unsatisfactory since we obtained a 78% retention rate, a percentage below the standards expounded by Harris et al. [39] and Walters et al. [40], with a pooled percentage of 80% and a total percentage of 89%, respectively. Therefore, the retention rate should be specifically taken into consideration by means of developing specific strategies to enhance the retention rates, as stated elsewhere [41].

Even though the application of the results stemming from a pilot study for effect size and sample size estimations should be cautiously considered [38], based on the values of the Cohen’s d (d = 1.17) obtained on the variable length of sway under the open-eyes condition for the experimental group (pre- to post-assessment), and considering a two-tailed hypothesis, the estimated sample size required would correspond to 26 subjects overall (13 subjects per study arm), which is somehow in line with the a priori size calculated (42 subjects).

## 5. Limitations

Although we have followed a rigorous methodological design, this study should be interpreted in light of its limitations. On the one hand, by including only one soccer team, the sample size was small, and we experienced a 22% patient loss rate after randomization. We could have increased the sample by including players from other teams, but we dismissed the idea after considering that different training routines would certainly affect the results. Therefore, the study corresponds to a pilot randomized controlled study and further research should focus on broader samples meeting the required minimal size. Moreover, our study was carried out with a sample of players from a club in a second division category; therefore, the results are not necessarily transferable to elite players or higher sport levels.

## 6. Conclusions

The results stemming from this pilot RCT have shown that an 8-week semi-supervised IMT program, performed with a threshold loading device, was not able to improve the static balance in a sample of soccer players. Moreover, no between-group differences were found regarding respiratory muscle function. Conducting a high-scale study is feasible and could refine the results and conclusions stemming from the current pilot study.

## Figures and Tables

**Figure 1 healthcare-11-00262-f001:**
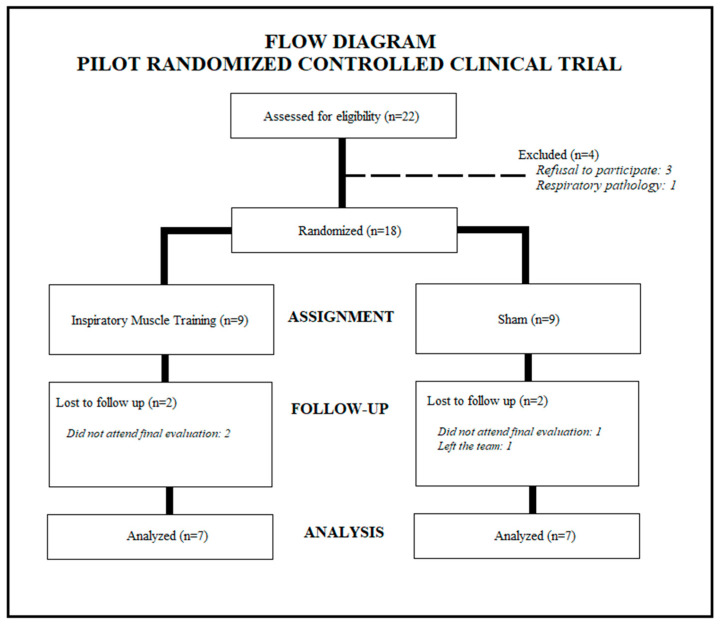
Participant selection and randomization process.

**Figure 2 healthcare-11-00262-f002:**
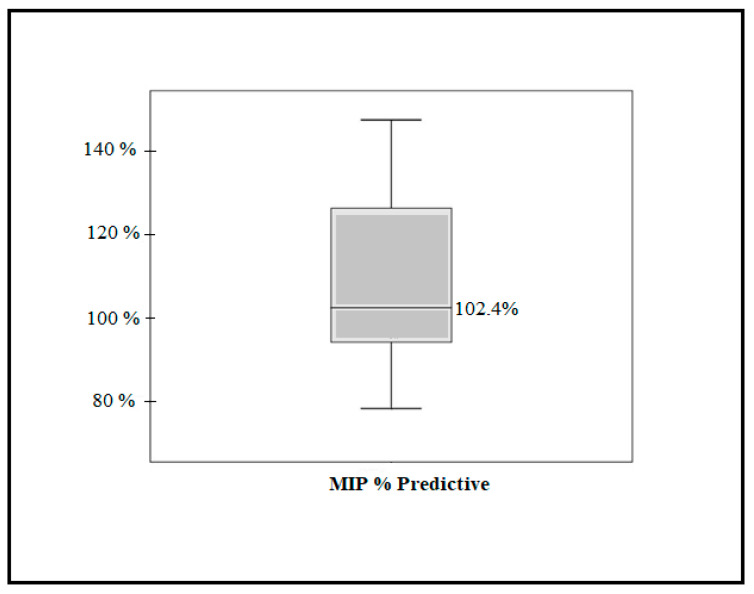
Baseline inspiratory muscle strength (MIP% predictive).

**Table 1 healthcare-11-00262-t001:** General characteristics of the participants (*n* = 14) at baseline.

Variables	IMT (*n* = 7)	Sham (*n* = 7)	*p*
Sociodemographic			
Age (years)	20.00 ± 0.81	20.00 ± 0.57	1.000
Anthropometric			
Size (cm)	177.57 ± 4.23	180.42 ± 5.79	0.313
Weight (kg)	71.88 ± 3.79	77.45 ± 8.70	0.147
Respiratory muscle function			
MIP (cm H_2_O)	161.57 ± 31.11	175.42 ± 30.27	0.415
MIP (% pred)	104.80 ± 21.4	111.50 ± 20.80	0.169
MVV (liters)	188.74 ± 30.06	201.78 ± 23.15	0.381
MVV (bpm)	89.89 ± 15.30	86.04 ± 21.58	0.707
MVV (% pred)	98.43 ± 14.00	103.39 ± 12.31	0.495
Static balance			
BOA_LS (mm)	2904.81 ± 640.03	4358.13 ± 1863.14	0.075
BOA_SE (mm^2^)	70.44 ± 42.04	123.14 ± 56.10	0.070
BOA_DX (mm)	10.06 ± 3.16	12.05 ± 4.00	0.322
BOA_DY (mm)	10.91 ± 3.47	14.75 ± 3.92	0.077
BOC_LS (mm)	3166.22 ± 641.30	4290.61± 1453.5	0.086
BOC_SE (mm^2^)	78.37 ± 63.52	127.21 ± 74.39	0.211
BOC_DX (mm)	10.01 ± 5.72	12.60 ± 2.85	0.305
BOC_DY (mm)	11.19 ± 4.41	17.43 ± 8.75	0.118

Data are expressed as mean ± standard deviation. MIP = maximum inspiratory pressure; % pred = % predictive; MVV = maximum voluntary ventilation; bpm = breaths per minute; BOA = bipodal eyes open; LS = length of sway; SE = surface of the ellipse; mm^2^ = square millimeters; DX = lateral axis; DY = antero-posterior axis; BOC = bipodal eyes closed.

**Table 2 healthcare-11-00262-t002:** Differences in static balance variables between groups pre–post-intervention.

	IMT (*n* = 7)				Sham (*n* = 7)				Differences within Interventions (Post-I–Pre-I)		Differences between Interventions (Post-I–Pre-I)
	Pre-IN	Post-IN	*p*	η^2^	Pre-IN	Post-IN	*p*	η^2^	IMT	Sham	IMT-Sham
BOA_LS (mm)	2904.8 ± 640.0	3522.4 ± 509.0	0.012	0.675	4358.1 ± 1863.1	3643.3 ± 596.7	0.286	0.186	617.6 ± 817.7	−714.8 ± 1956.2	1332.4 (−787.5, 3452.3)
BOA_SE (mm^2^)	70.4 ± 42.0	177.5 ± 171.3	0.175	0.282	123.1 ± 56.1	162.8 ± 115.9	0.375	0.133	107.1 ± 176.3	39.7 ± 128.7	67.4 (−150.6, 285.6)
BOA_DX (mm)	10.0 ± 3.1	12.8 ± 5.3	0.220	0.238	12.0 ± 4.0	15.1 ± 7.7	0.329	0.159	2.8± 6.1	3.1 ± 8.6	−0.3 (−10.9, 10.3)
BOA_DY (mm)	10.9 ± 3.4	12.3 ± 3.4	0.090	0.405	14.7 ± 3.9	18.4 ± 8.1	0.223	0.235	1.4 ± 4.8	3.7 ± 8.9	−2.3 (−12.4, 7.8)
BOC_LS (mm)	3166.2 ± 641.3	4173.3 ± 390.8	0.004	0.779	4290.6 ± 1453.5	4006.5 ± 787.1	0.554	0.062	1007.1 ± 750.7	−284.0 ± 1650.4	1291.2 (−521.9, 3104.3)
BOC_SE (mm^2^)	78.3 ± 63.5	78.4 ± 51.2	0.998	0.000	127.2 ± 74.3	120.8 ± 127.7	0.888	0.004	0.1 ± 81.6	−6.4 ± 147.7	6.5 (−162.2, 175.2)
BOC_DX (mm)	10.0 ± 5.7	12.8 ± 5.3	0.338	0.153	12.6 ± 2.8	15.1 ± 7.7	0.341	0.152	2.7 ± 7.8	2.5 ± 8.2	0.3 (−11.0, 11.6)
BOC_DY (mm)	11.1 ± 4.4	12.3 ± 3.4	0.370	0.135	17.4 ± 8.7	18.4 ± 8.1	0.808	0.011	1.2 ± 5.5	1.0 ± 11.9	−2.2 (−15.4, 11.0)

Data are expressed as mean ± standard deviation or mean (95% confidence interval). Pre-IN: pre-intervention; Post-IN: post-intervention; η^2^ = effect size. BOA = bilateral eyes open; LS = length of sway; SE = surface of the ellipse; mm^2^ = squared millimeters; DX = lateral axis; DY = antero-posterior axis; BOC = bilateral eyes closed.

**Table 3 healthcare-11-00262-t003:** Differences in respiratory function variables between groups pre–post-intervention.

	IMT (*n* = 7)				Sham (*n* = 7)				Differences within Interventions (Post-I–Pre-I)		Differences between Interventions (Post-I–Pre-I)
	Pre-I	Post-I	*p*	η^2^	Pre-I	Post-I	*p*	η^2^	IMT	Sham	IMT-Sham
MIP (cm H_2_O)	161.5 ± 31.1	184.4 ± 21.5	0.076	0.434	175.4 ± 30.2	176.0 ± 16.4	0.951	0.001	22.9 ± 37.8	0.6 ± 34.3	22.3 (−28.74, 73.34)
MIP (% pred)	104.8 ± 21.4	119.5 ± 14.5	0.076	0.432	111.5 ± 20.8	111.6 ± 10.5	0.981	0.000	14.7 ± 25.8	0.1 ± 23.3	14.6 (−20.1, 49.3)
MVV (liters)	188.7 ± 30.0	169.5 ± 37.9	0.180	0.278	201.7 ± 23.1	197.2 ± 26.6	0.579	0.054	−19.2 ± 48.3	−4.5 ± 35.2	−14.7 (−74.4, 45.0)
MVV (bpm)	89.8 ± 15.3	143.2 ± 21.9	0.005	0.756	86.0 ± 21.5	147.8 ± 33.4	0.000	0.890	53.4 ± 26.7	61.8 ± 39.7	−8.4 (−56.2, 39.4)
MVV (% pred)	98.4 ± 14.0	88.4 ± 19.0	0.187	0.270	103.3 ± 12.3	101.2 ± 15.6	0.611	0.046	−10.0 ± 23.6	−2.1 ± 19.8	−7.9 (−38.7, 22.9)

Data are expressed as mean ± standard deviation or mean (95% confidence interval). Pre-I = pre-intervention; Post-I = post-intervention; MIP = maximum inspiratory pressure; % pred = % predictive; MVV = maximum voluntary ventilation; bpm= breaths per minute; η2 = effect size.

## Data Availability

The study was registered on ClinicalTrials.gov (accessed on 26 November 2022) under the following ID: NCT03383900.

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
