# Peer review of "Does Inspiratory Muscle Training Affect Static Balance in Soccer Players? A Pilot Randomized Controlled Clinical Trial"

_healthcare, 2023, doi:10.3390/healthcare11020262_

Round 1
Reviewer 1 Report
I have been asked to review this manuscript before and I still have serious concerns about the concepts described and the design followed to conduct this particular study. In my opinion the results of this study demonstrate there is no actual connection between the intervention administered, the population selected and the outcome measures used.
Firstly, the population selected had no apparent need for indirect balance training (via respiratory strengthening exercise). There were no apparent balance deficits to be addressed nor any injury risk factors that could be lowered with respiratory strengthening. Why would an apparently healthy population set of young individuals require additional respiratory training and why would this type of training affect the diaphragm’s contribution to the balance ability of the whole body? Lastly, why was the Mini BESTest used as a pivotal outcome for balance assessment, as the population it was intended for were patients with neurological (stroke, Parkinson’s, Multiple Sclerosis, cerebellar ataxia etc.) or vestibular deficits? Please see here: https://www.bestest.us/
The authors attempt, inappropriately in my opinion, to bring the current research paradigm into context by referring to very particular situations in soccer and NOT whole body balance but lumbar spine motor control for inter-segmental stability issues (see lines 48-49): “when there is either weakness and/or an increase in the respiratory demand (i.e. in a soccer match), the key role of the diaphragm in the stability of the lumbar region declines, especially due to early muscle fatigue”. The case-control study referred immediately after (lines 51-53) could be an indication for performing this research, although not convincing as it is from a single study, and most likely for now it can be considered as circumstantial evidence at best, as the 2 populations compared could have differed in many other factors not measured.
Lines 254-257 are guidelines on how to write the discussion provided by ‘Healthcare’, not to be used as an introductory paragraph of the Discussion section. If we read further, the authors declare their surprise to the results obtained (line 269) and that these should be researched further (or not at all, in my opinion), as comparing the results of this study to other dissimilar studies is rather unacceptable.
Author Response
Dear Reviewer 1: We have enclosed the itemized response letter in enclosed file. Please see the attachment. Thank you very much.

Reviewer 2 Report
Dear Authors,
I would like to express my gratitude regarding the opportunity to review this manuscript.
At this stage the document requires improvements, below with line indication:
6-15 – Please correct considering the journal template and journal instructions for authors.
17-30 – Please consider increasing the quality of the abstract. Readers should clearly understand the study.
33-60 – Please consider developing the introduction section, namely characterizing and relating the sport with the topic under study (IMT).
67 – Please indicate ID number.
72-78 - Please describe in detail the inclusion and exclusion criteria.
72-78 – Please describe the sample and characteristics (height, body mass, and others).
75 – “Exclusion criteria included” – Please revise the English in this line, and throughout the manuscript.
81 – “Sham” or “control group”? Please consider and describe what is more appropriate.
83 – Please describe the used equipment’s and characterize them (for example with the city and country of the manufacturer).
95-99 – Please provide scientific reference to support the procedure.
104 - Please describe who collected the data (academic background and experience), and other details related to data collection (time of day, temperature, humility, places for data collection. and others). Please also address the familiarization process previously to data collection.
141-151 – Please include normality test, sample power, and description of data presented with mean and SD.
168 – “et al.” – Please correct. Not only in this line, but in many other example throughout the manuscript.
172 – Figure description usually below the figure, please review journal template and journal instructions for authors.
173 – Please improve the figure quality.
179 – Please remove the space.
180-189 – Please format the table according to the journal template and journal instructions for authors. Please also correct the table footnote format.
216-220 – Please remove the lines, aiming table and footnote in the same page.
Table 2 – Please format the page, please format the table. For example, all results in the same column.
230-232 – Please improve the figure quality and review the table description place in the page, such as in figure 1, considering the journal template and journal instructions for authors.
234 , 242, 243 – Please delete.
Table 3 - 180-189 – Please format the table according to the journal template and journal instructions for authors. Please also correct the table footnote format.
254-257 – Please remove.
268-293 – Please consider dividing the paragraph, it is too long.
315 – “:” – does not seem adequate. Please review.
344-348 – This section should be more developed, with clear take-home messages and possible practical application (which should also be considered in the end of the abstract in page 1).
368 – Please insert space.
374-351 - All references should be carefully revised, they are not according to the journal template and journal instructions for authors.
Please carefully revise the English throughout the manuscript.
Author Response
Dear Reviewer 2: We have enclosed the itemized response letter in enclosed file. Please see the attachment. Thank you very much.

Reviewer 3 Report
The manuscript is an attempt to evaluate the effect of inspiratory muscle training on the postural stability and balance in soccer players. The introduction of the manuscript contains the necessary information to justify and convince in the importance of the study. The section materials and methods is described in details and generally the methodological approach is correct. However, the authors state that the study has been carried out from February to May ( three months) and at the same time the trainings lasted 8 weeks? They should explain better this difference in time. The studied variables that are supposed to be influenced by IMT are properly selected. The statistical procedures are correct. The results are clearly presented and illustrated with adequate tables and figures. The discussion is skilful. A specific remark, more a technical issue: the first paragraph of the discussion explaining the way of discussing should be removed. The conclusion are sound and derived from the results. The only shortcoming of the study, in my opinion is the rather small sample size, however, the authors have made the necessary verifications to classify this as a pilot study, explained very well the limitations in regard to the significance of the results . In their future work, however, I will recommend to use larger samples, above the set limits. In my opinion,despite the minor corrections needed the manuscript has a good merit, fits entirely into the scope of Healthcare journal and might be accepted after corrections.
Author Response
Dear Reviewer 3: We have enclosed the itemized response letter in enclosed file. Please see the attachment. Thank you very much.

Round 2
Reviewer 1 Report
I am glad the authors recognized my comments' essence and addressed them appropriately. The manuscript is sufficiently transformed to be able to proceed to publication. Thank you.
Author Response
Dear Reviewer 1: Thank you very much for your kindness and for setting your
expertise at our manuscript’s disposal in order to enhance and improve the general quality of our paper. Thank you for your acceptance.
Reviewer 2 Report
Dear Authors,
Thank you for considering my suggestions and incorporating them into the manuscript, which globally improved, congratulations.
Below suggestions related to this last version (v2), with line indication.
4-12 - Please consider the journal template and instructions for authors regarding the manuscript format.
15-16 – Please clearly indicate the study aim and characterize the sample.
28 – Please remove end point.
32-33 – “demands” and “demand”. Please consider improving the English throughout the manuscript, a careful analysis is suggested. For example, the same in line 88 and in lines 120-124 – these are only some examples.
101 and others – Please review the template considering the format of subtopics.
123 – “UCAM” was previously abbreviated in the manuscript.
161 –It is suggested to indicate in this section that the results are presented with mean and SD.
180-186 – This text regarding sample power is suggested in line 161.
192 – Please review the figure content: “sham”? Some text with uppercase and in other with lowercase, please standardize.
193-194 – Please reformulate considering the journal template and instructions for authors.
211-213 - Please reformulate considering the journal template and instructions for authors.
214 – Please remove the space.
226 – “stated”? “observed” or “found”? Please carefully revise the English throughout the manuscript, this is very important.
240 – Please consider improving the reading conditions of table content, for example the “±” should be standardized. “Sham” should be previously described. Please also consider in table 3.
241 - Please reformulate considering the journal template and instructions for authors. Please also consider in table 3.
252 - Please reformulate considering the journal template and instructions for authors.
370 – Please consider developing the conclusions.
384 – “and” is missing.
394 – Please insert space.
401 – References are not according to journal template and instructions for authors. Corrections are very important.
Author Response
Dear Reviewer 2: Thank you very much for your indications. We have enclosed an itemized point-by-point response letter. Please see attachment. Thank you.
